# EpCAM silencing suppresses aggressive phenotypes and induces partial redifferentiation in anaplastic thyroid cancer cells

**Teruo Nakamura☯, Tomohiro Shibata☯, Ken-ichi Ito✉*

Division of Breast and Endocrine Surgery, Department of Surgery, Shinshu University School of Medicine, Asahi, Matsumoto, Japan

☯ N.T. and S.T. contributed equally to this work.
* kenito@shinshu-u.ac.jp

## Abstract

Anaplastic thyroid cancer (ATC) is a rare but highly aggressive malignancy with a dismal prognosis. Although recent advances in targeted therapies have modestly improved survival, the molecular mechanisms driving ATC progression remain incompletely elucidated. Epithelial cell adhesion molecule (EpCAM), a multifunctional cell-surface protein, is implicated in proliferation, migration, and stemness in various cancers. However, its role in thyroid cancer progression remains unclear. In this study, we investigated the function of EpCAM in thyroid cancer cell lines of varying differentiation status. EpCAM expression was significantly elevated in ATC cell lines compared with differentiated thyroid cancer (DTC) lines. EpCAM knockdown by siRNA suppressed proliferation, adhesion, motility, and invasion in ATC cells, but had minimal effects on DTC cells. Morphological analyses revealed that EpCAM silencing induced differentiation features, including follicle-like structure formation and increased expression of thyroid differentiation markers such as thyroglobulin and PAX8 in ATC cells. Furthermore, EpCAM inhibition decreased mesenchymal marker expression, reduced filopodia formation, and suppressed extravasation of cancer cells into the lung in an *in vivo* mouse model. Mechanistically, EpCAM knockdown attenuated epithelial–mesenchymal transition (EMT)-related pathways but did not affect major proliferation signaling cascades in ATC cells. These findings suggest that EpCAM promotes dedifferentiation and metastatic potential in ATC through EMT modulation. Our results provide new insights into the role of EpCAM in thyroid cancer biology and highlight its potential as a therapeutic target in ATC. Further studies are warranted to elucidate the mechanisms linking EpCAM to anaplastic transformation and to explore the therapeutic efficacy of EpCAM-targeting strategies in aggressive thyroid cancers.

**Data availability statement:** All relevant data are within the manuscript and its Supporting Information files.

**Funding:** This work was supported by a Grant-in-Aid for Scientific Research from the Japan Society for the Promotion of Science. KI:#20K08953 TS:#20J00393 and #21K06641.

**Competing interests:** The authors have declared that no competing interests exist. TN is currently an employee of Chugai Pharmaceutical Co., Ltd., but has no conflict of interest related to this study.

## Introduction

Anaplastic thyroid cancer (ATC) is a small subset of thyroid cancers that is nearly incurable, with a median survival of approximately 6 months [1,2]. Although ATC accounts for only 2–3% of all thyroid cancers, it is responsible for approximately 50% of thyroid cancer-related deaths [3,4]. In recent years, the introduction of inhibitors targeting BRAF, RET, and NTRK, as well as immune checkpoint inhibitors, has extended the survival time of patients with ATC; however, not many patients survive for more than 2 years [2].

Thyroid cancer is generally thought to arise from the transformation of normal thyroid follicular epithelial cells into differentiated thyroid carcinoma (DTC), with subsequent progression through the stepwise accumulation of genetic and epigenetic alterations, leading to poorly differentiated thyroid cancer (PDTC) or highly aggressive ATC [5]. However, accumulating evidence suggests that not all ATCs necessarily originate from a pre-existing differentiated component. Genomic analyses have demonstrated that ATC and papillary thyroid cancer (PTC) may diverge early during tumor evolution and can evolve as molecularly distinct entities, even in cases where both components coexist within the same tumor [6]. In addition to driver mutations commonly identified in DTC, ATC exhibits a high frequency of alterations in genes such as the *TERT* promoter and *TP53* [7]. Nevertheless, the molecular mechanisms underlying the emergence of ATC, including both dedifferentiation from DTC and alternative evolutionary pathways, remain incompletely understood [8].

Epithelial cell adhesion molecule (EpCAM) is a member of the immunoglobulin cell adhesion molecule family. Its functions include cell signaling, differentiation, migration, adhesion, and proliferation [9]. Its cleaved intracellular domain enters the nucleus, leading to enhanced activation of β-catenin, c-Myc, and cyclin D1, thereby promoting cancer cell proliferation [10]. We have previously reported that the expression levels of EpCAM and CD44v6 are markedly increased in ATC cell lines and clinical specimens [11]. EpCAM has been implicated in carcinogenesis and is expressed in various cancer types, including liver, lung, breast, ovarian, gastric, and colorectal cancers [12–17]. Furthermore, EpCAM-overexpressing carcinoma cells exhibit stem cell–like phenotypes, resulting in high rates of recurrence, metastasis, and drug resistance [16]. Taken together, these findings suggest that EpCAM actively participates in the progression of several cancers; however, its function and role in thyroid cancer progression have not yet been elucidated.

In this study, we explored whether EpCAM is involved in cellular activities related to the emergence of the malignant phenotype of ATC. Our findings not only shed light on the molecular mechanisms underlying EpCAM-mediated dedifferentiation and the acquisition of malignant potential in thyroid cancer, but also demonstrate that EpCAM may serve as a therapeutic target against aggressive thyroid cancer.

## Materials and methods

### Cell lines and chemicals

TPC-1 cells, originating from PTC, KTC-1 cells from PDTC, and FRO cells from ATC were provided by Dr. Yamashita of Nagasaki University [18]. FTC-133 and WRO

cells, which originate from follicular thyroid cancer (FTC), were provided by Dr. Takeda of Shinshu University [19]. ACT-1 cells, derived from ATC, were provided by Dr. Ohara of Tokushima University [20]. OCUT-1F and OCUT-3 cells, both derived from ATC, were provided by Dr. Onoda of Osaka City University [21]. Cells were cultured at 37 °C in Roswell Park Memorial Institute (RPMI) medium supplemented with 10% fetal bovine serum (FBS). The cell lines used in this study were authenticated by short tandem repeat profiling. All the experiments were performed with mycoplasma-free cells. The characteristics of the cell lines used in this study are listed in S1 Table.

### Small interfering RNA (siRNA) transfection

The siRNA corresponding to the nucleotide sequence of EpCAM was purchased from Invitrogen (Carlsbad, CA, USA). Cells were transfected with EpCAM siRNA 1 (10 nM; Assay ID: 11358) or EpCAM siRNA 2 (10 nM; Assay ID: 11452) using Lipofectamine RNAiMAX and Opti-MEM (Invitrogen), according to the manufacturer's instructions. In all experiments, EpCAM siRNA–transfected cells were seeded 24 h after transfection.

### Colony formation assay

Cells transfected with EpCAM siRNA (10 nM) or control siRNA (10 nM) for 24 h were seeded in 24-well plates ($1.5 \times 10^3$ cells per well) and cultured in RPMI medium supplemented with 10% FBS for 7 days. The cells were then washed once with $1 \times$ phosphate-buffered saline (PBS), fixed with methanol, and stained with 0.5% crystal violet.

### Cell adhesion assays

In the Matrigel adhesion assay, 48-well plates were precoated with 50 mg·mL$^{-1}$ Matrigel (BD Biosciences, USA) for 6 h at 37 °C. Then, EpCAM siRNA–transfected cells ($1.0 \times 10^4$ cells per well) were seeded into the plates and incubated at 37 °C for 30 min. Nonadherent cells were washed thrice with PBS, and cells that adhered to the Matrigel were counted from three randomly selected fields per well using an inverted microscope (Olympus Corp. Tokyo, Japan) at $40 \times$ magnification. Independent experiments were performed in triplicate. For the ECM adhesion assay, EpCAM siRNA–transfected cells ($1.0 \times 10^4$ cells per well) were seeded into a 48-well cell adhesion assay kit (CBA-070; Cell Biolabs, Inc., CA, USA) and incubated at 37 °C for 30 min. The cells were stained according to the manufacturer's recommendations, and absorbance at 560 nm was measured using a microplate reader (Agilent Technologies, Santa Clara, CA, USA)

### Quantitative reverse transcription polymerase chain reaction (qRT-PCR)

Total RNA was isolated from cells using ISOGEN (311−02501; Nippon Gene Co., Ltd., Tokyo, Japan) according to the manufacturer's instructions. RNA (1 µg) was reverse-transcribed using random hexamers and AMV reverse transcriptase (Promega, Madison, WI). qRT-PCR was performed using the Real-Time PCR system 7300 (Applied Biosystems, Foster City, CA) in reaction mixtures containing 1 µL cDNA, primer pairs, a dual-labeled fluorogenic probe, and TaqMan Universal PCR Master Mix (Applied Biosystems, Santa Clara, CA). TaqMan Gene Expression Assays for EpCAM, thyroglobulin, sodium–iodide symporter (NIS), paired-box gene 8 (PAX8), thyroid transcription factor-1 (TTF-1), and β-actin were purchased from Applied Biosystems (Carlsbad, CA, USA). The thermal cycling conditions were 95 °C for 10 min, followed by 40 cycles of 95 °C for 15 s, and 60 °C for 1 min. Relative gene expression in each sample was determined using the following formula: $2^{-\Delta Ct} = 2^{[Ct (\beta\text{-actin}) - Ct (target)]}$. Target-gene expression was normalized to β-actin levels.

### Western blot analysis

Proteins were isolated from cells, as previously described and used for western blot analysis (10 µg/lane) [22,23]. Briefly, cells were rinsed with ice-cold PBS and lysed in buffer (pH 8.0) comprising 50 mM Tris-HCl, 250 mM NaCl, 0.3% NP-40, 1 mM EDTA, 10% glycerol, 0.1 mM Na$_3$VO$_4$, 50 mM NaF, 1 mM phenylmethylsulfonyl fluoride, 10 µg/mL

aprotinin, and 10 µg/mL leupeptin. The cell lysates were subjected to SDS-PAGE and transferred to Immobilon membranes (Millipore, Bedford, MA, USA). Membranes were then incubated with blocking solution followed by primary antibodies. Antibody detection was performed using Chemi-Lumi One L (Nacalai Tesque Inc., Japan). Luminescence intensity was quantified using a ChemiDoc™ MP Imaging System (Bio-Rad Laboratories, Tokyo, Japan). The antibodies used in this study are listed in S2 Table.

### Transwell migration assay

The migration assay was performed using a multiwell chamber as the outer chamber and 8 µm pore-size polycarbonate filters (BD Biosciences, Bedford, MA, USA) as the inner chamber. Transfected cells were seeded in the upper chambers of a 24-well plate ($5.0 \times 10^5$ cells per well) with 1% FBS RPMI, and RPMI medium with 10% FBS was added to the lower chambers. After 24 h of incubation, cells on the lower surface were fixed with methanol, stained with Giemsa, and counted under a microscope.

### Wound healing migration assay

The transfected cells were seeded into 24-well plates ($5.0 \times 10^4$ cells per well). Upon reaching confluence, the cell monolayer was scratched and maintained in RPMI medium. After incubation, images were captured using an inverted microscope ($40 \times$ magnification).

### Transwell invasion assay

The 8 µm pore-size polycarbonate filters were coated with Matrigel for 4 h at 37 °C. After transfection with EpCAM siRNA, cells were seeded into the upper chambers with RPMI medium containing 1% FBS. The lower chambers contained complete culture medium (RPMI supplemented with 10% FBS). After 24 h of incubation, cells on the lower surface were fixed with methanol, stained with Giemsa, and counted under a microscope.

### Morphological changes on Matrigel

Chilled Matrigel (200 µL) was used to coat each well of a 12-well plate and was allowed to polymerize at 37 °C for 2 h. Cells ($5.0 \times 10^4$ cells per well) were seeded onto the Matrigel. After 1, 4, 8, and 24 h of incubation, cell morphology was analyzed.

### Mice

All animal experiments were approved by the Animal Experiments Committee of Shinshu University (approval number: 020102) and were conducted in accordance with the recommendations of the United States Public Health Service Policy on Humane Care and Use of Laboratory Animals (Office of Laboratory Animal Welfare, NIH, Department of Health and Human Services, Bethesda, MD, USA). All animals were monitored daily by trained research team members and animal laboratory staff for signs of dehydration and distress. The mouse colonies (siControl, n = 6; siEpCAM, n = 6) were maintained at the Division of Animal Research, Shinshu University.

### *In vivo* assessment of extravasation of cancer cells into the lung

Male BALB/c nu/nu athymic nude mice (6–7 weeks old) were purchased from Charles River Laboratories (Yokohama, Japan). EpCAM siRNA–transfected ACT-1 cells were labeled with Cell Tracker™ Green CMFDA (C7025; Thermo Fisher Scientific, Waltham, MA, USA) for 24 h before injection, and the labeled cells suspended in PBS were injected into the tail vein ($1 \times 10^6$ cells per mouse). At 24 h post-injection, mice were anesthetized with a mixture of medetomidine (0.3 mg/kg), midazolam (4.0 mg/kg), and butorphanol (5.0 mg/kg) and perfused with PBS to flush cancer cells present in the

vasculature. Following PBS perfusion, mice were euthanized by cervical dislocation, and the lungs were subsequently harvested. At the end of the experiments, the lungs were perfused with PBS, after which half of the lung tissue was embedded in optimal cutting temperature compound (Sakura Fintek Japan Co., Ltd, Japan) for cryosectioning, and the remaining half was analyzed via flow cytometry.

### Flow cytometry

Lung tissues were minced in PBS containing collagenase A (10103586001; Roche Diagnostics GmbH, Mannheim, Germany) and deoxyribonuclease I (LS002138; Worthington Biochemical, Lakewood, NJ, USA) at final concentrations of 0.5% and 20 U/mL, respectively. The mixture was incubated for 1 h at 37 °C with gentle agitation. Digestion was stopped by the addition of FBS, after which the cell suspension was washed with PBS and passed through a 100 μm mesh nylon screen. Dead cells were stained using Zombie UV™ Fixable Viability Kit (BioLegend, San Diego, CA, USA). Both forward scatter (FSC) and side scatter (SSC) parameters were used for analysis. A BD FACSCanto™ II system (Becton-Dickinson, NJ, USA) was used to detect the labeled cells. Data analysis was conducted using FlowJo software, and flow cytometry data are presented as histograms and dot plots.

### Statistical analysis

Experimental results are expressed as the mean ± SD or SEM. Statistical differences between groups were assessed using a two-tailed Student's $t$ test. For comparisons involving multiple groups, one-way or two-way ANOVA followed by Bonferroni's correction was applied for all pairwise comparisons. A $p$-value < 0.05 was considered statistically significant. Statistical analyses were conducted using GraphPad Prism 10 software.

## Results

### EpCAM silencing inhibits cell growth and motility in anaplastic thyroid carcinoma cell lines

We examined the protein expression levels of growth- and motility-related genes in eight human thyroid cancer cell lines using western blot analysis. As shown in Fig 1A, the expression levels of EpCAM and claudin 7 were highest in the ATC cell lines FRO and ACT-1, which is consistent with the results of our previous study [11]. In ATC cell lines, the MEK/ERK signaling pathway was activated compared with that in cells of other subtypes, except for KTC-1. In contrast, AKT signaling was highly activated in the FTC cell line FTC-133, which harbors a *PTEN* mutation. Phosphorylated β-catenin was more prominently detected in KTC-1 and FTC-133 cells compared with the other thyroid cancer cell lines. In contrast, MMP2 expression showed relatively minor differences among the cell lines examined. E-cadherin expression was detected only in the ATC cell lines FRO and ACT-1, and these two cell lines also expressed N-cadherin and multiple mesenchymal marker proteins. In the other thyroid cancer cell lines, E-cadherin expression was not detected, whereas the expression of multiple mesenchymal markers was observed. FRO and ACT-1 cells were found to express EpCAM at significantly higher levels than other thyroid cancer cell lines. Accordingly, these two cell lines were primarily used in subsequent experiments to analyze alterations in cellular behavior following EpCAM suppression.

Next, we examined whether EpCAM affected the proliferation of thyroid cancer cells. Treatment with EpCAM siRNA significantly decreased the proliferation of the high EpCAM-expressing ATC cell lines FRO and ACT-1 ($p$ < 0.01) (Fig 1B). Moreover, in the clonogenic assay, EpCAM knockdown significantly decreased colony numbers in FRO and ACT-1 cells but did not affect the colony number in the low EpCAM–expressing TPC-1 and FTC-133 cells (Fig 1C).

As EpCAM is associated with the mesenchymal phenotype of cancer cells [24], we further examined whether it affects cell motility using a transwell migration assay (Fig 1D) and scratch wound healing assay (Fig 1E). EpCAM knockdown significantly reduced the number of migrated FRO and ACT-1 cells (Figs 1D and E).

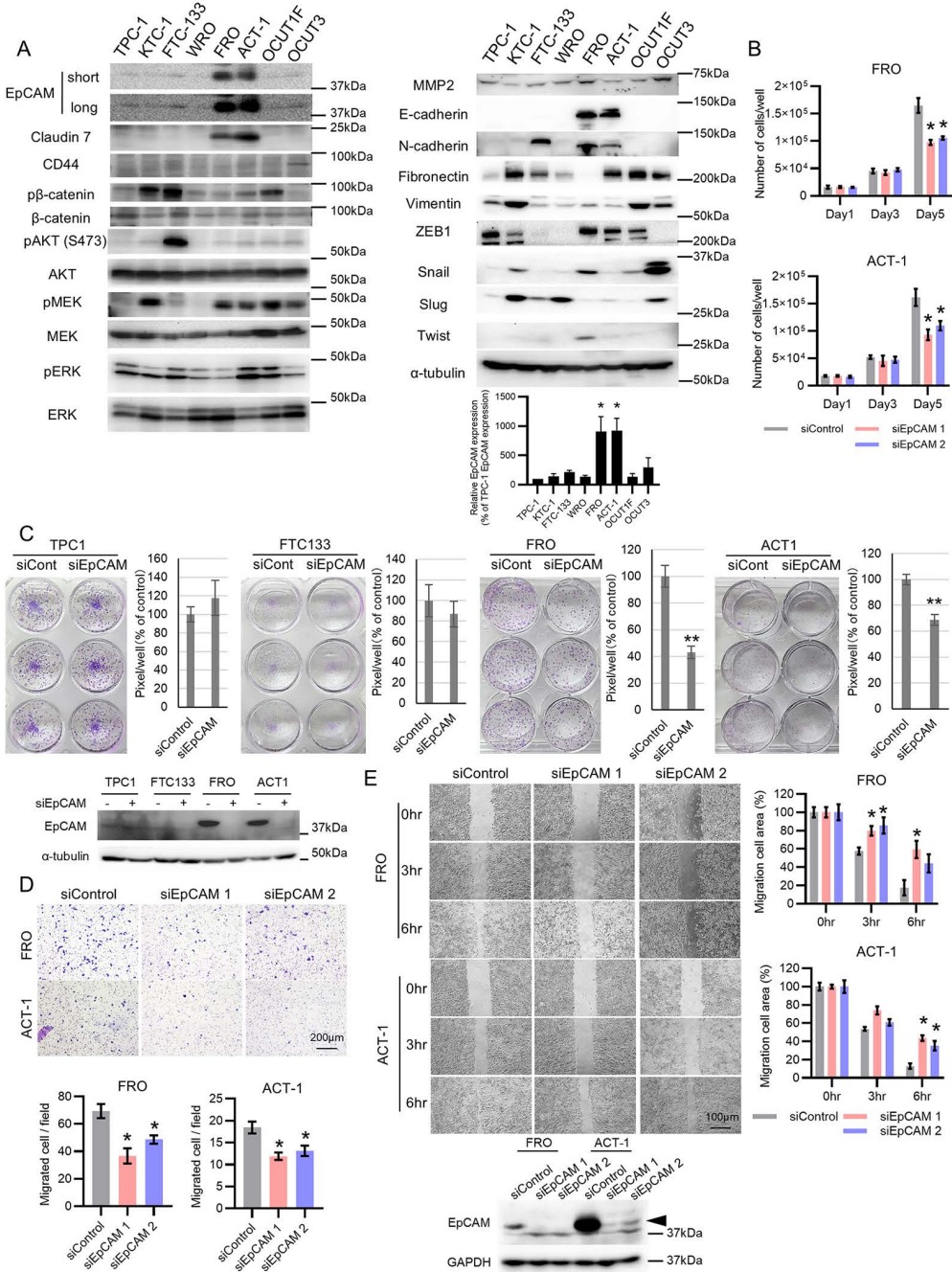

**Fig 1. Effect of EpCAM knockdown on cell proliferation and motility in thyroid cancer cell lines. A**. Western blot analysis showing the expression of EpCAM, cell survival signaling molecules, and cell motility–related molecules in eight thyroid cancer cell lines. "short" indicates the blot obtained after short exposure time, while "long" indicates results from a long exposure time of the same western blot (upper panel). EpCAM expression levels were normalized to α-tubulin, and relative expression levels compared with TPC-1, calculated from three independent experiments, are shown in the bar charts (lower panel). Data are presented as mean±SD (*p<0.05 vs. TPC-1). **B**. Growth rates of FRO and ACT-1 cells assessed after treatment with EpCAM siRNA for 1, 3, or 5 d. *p<0.01. **C.** Colony formation assay of TPC-1, FTC-133, FRO, and ACT-1 cells transfected with control or EpCAM siRNA. Cells were treated with EpCAM siRNA (10 nM) for 24 h, after which the medium was replaced and cells were cultured for an additional 7 d. The colonies were stained with crystal violet (upper panel). Western blot analysis showing the effect of EpCAM knockdown (lower panel). **D.** Transwell migration assay of FRO and ACT-1 cells transfected with control or two independent EpCAM siRNAs. The number of migrated cells is shown as a bar chart. **E**. Scratch wound healing assay of FRO and ACT-1 cells transfected with control or two independent EpCAM siRNAs. Representative images of the scratch

wound assay at 0, 3, and 6 h after scratching the confluent cell monolayer (left panel). The gap length was measured for each sample, and the average ratio of the residual gap to the initial gap is shown in the bar chart (right panel). Western blot analysis showing the effect of EpCAM knockdown (lower panel). Data are presented as the mean ± S.D. n = 3. *$p < 0.05$, **$p < 0.01$.

## EpCAM silencing inhibits cell adhesion and invasion in anaplastic thyroid carcinoma cell lines

As EpCAM expression is closely associated with cell motility, we examined whether EpCAM affects thyroid cancer cell invasion. First, we examined its effect on the adhesion of four thyroid cancer cell lines to the extracellular matrix using a cell adhesion assay. EpCAM knockdown significantly reduced the adhesion of TPC-1, FRO, and ACT-1 cells to Matrigel, but not that of FTC-133 cells (Fig 2A).

To identify the extracellular matrix components that adhere to EpCAM, we assessed the ability of FRO and ACT-1 cells to adhere to each substrate using a cell adhesion assay kit. EpCAM knockdown resulted in significantly decreased adhesion of both cell lines to collagen IV- and laminin I-coated plates (Fig 2B). As collagen IV and laminin I constitute major components of Matrigel, these results suggest that these cells can adhere to Matrigel through interactions between EpCAM and collagen IV or laminin I. Furthermore, when the invasive ability of TPC-1, FTC-133, FRO, and ACT-1 cells was examined using a Transwell invasion assay, EpCAM knockdown significantly decreased the invasion of FRO and ACT-1 cells but not of TPC-1 and FTC-133 cells (Fig 2C).

## EpCAM silencing inhibits dedifferentiation of anaplastic thyroid carcinoma cells

As EpCAM is highly expressed in ATC cell lines, we further investigated whether it is involved in the dedifferentiation of DTC cells. We first examined the morphogenesis of thyroid cancer cells on Matrigel, as we previously observed that immortalized normal thyroid follicular epithelial cells form thyroid follicle–like structures on Matrigel within a short time [25]. DTC cell lines (TPC-1, KTC-1, FTC-133, and WRO) formed follicle–like network structures within 12 h of seeding on Matrigel (Fig 3A). In contrast, ATC cell lines (FRO, ACT-1, OCUT-1F, and OCUT-3) did not form follicle-like network structures, and ACT-1 and OCUT-1F cells aggregated in a time-dependent manner. However, when EpCAM expression was inhibited by siRNA, network formation was observed in ATC cell lines (FRO and ACT-1), whereas no changes in cellular morphogenesis were observed in the DTC cell lines (TPC-1 and FTC-133) (Fig 3B).

Since EpCAM is highly expressed in ATC cell lines, we next examined whether EpCAM silencing affects the expression of thyroid differentiation markers in DTC and ATC cell lines seeded on Matrigel. EpCAM knockdown significantly increased the expression of thyroglobulin, PAX8, and TTF-1 in the ATC cell line FRO (Fig 3C). In contrast, EpCAM knockdown did not alter the expression of these differentiation markers in DTC cell lines, with the exception of NIS in FTC-133 cells. These findings indicate that EpCAM silencing is associated with increased expression of thyroid differentiation markers in ATC cells with high EpCAM expression.

## EpCAM silencing inhibits filopodia formation in anaplastic thyroid carcinoma cell lines

To identify key molecules involved in EpCAM-induced invasion and metastasis, we examined the expression levels of cell growth- and motility-related genes in cells transfected with EpCAM siRNA. Fig 4A shows alterations in the expression of molecules related to signal transduction, cell growth, and epithelial–mesenchymal transition (EMT) in thyroid cancer cells treated with EpCAM siRNA. EpCAM knockdown resulted in a marked reduction of AKT phosphorylation in TPC-1 and FTC-133 cells, with an approximately tenfold decrease compared with control siRNA-treated cells. In contrast, a modest increase in phospho-AKT levels was observed in the ATC cell lines FRO and ACT-1. Notably, the magnitude of AKT phosphorylation changes in FRO and ACT-1 cells was considerably smaller than the pronounced decrease observed in TPC-1 and FTC-133 cells, and was not accompanied by consistent changes in other major cell growth–related signaling molecules (Fig 4A). In addition, EpCAM knockdown decreased the expression of the mesenchymal markers Snail and Slug in

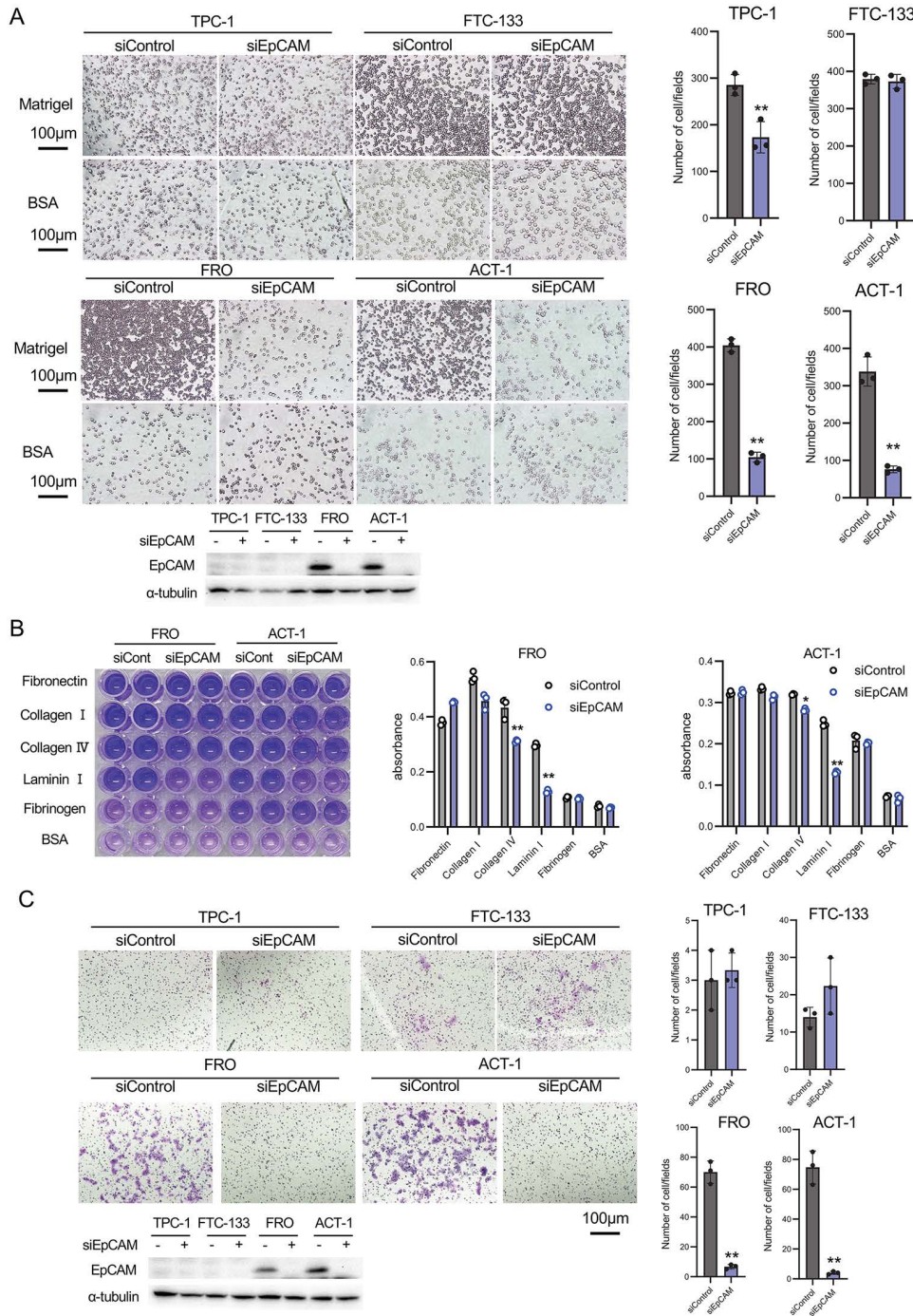

**Fig 2. Effect of EpCAM knockdown on cell adhesion and invasion in thyroid cancer cell lines. A.** Cell adhesion assay using BSA- or Matrigel-coated plastic dishes. Representative images are shown in the left panel, and quantification of cell numbers is shown in the right panel. **$p < 0.01$. **B.** Cell adhesion assay using ECM protein–coated plates. Adherent cells were stained (left panel) and quantified by measuring absorbance at 560 nm after extraction (right panel). *$p < 0.05$, **$p < 0.01$. **C**. Transwell invasion assay for TPC-1, FTC-133, FRO, and ACT-1 cells transfected with EpCAM siRNA. Cells that invaded the Matrigel-coated Transwell chamber were counted as described in the Methods section. The number of migrated cells is shown as a bar chart. Data are presented as the mean ± S.D., n = 3. **$p < 0.01$.

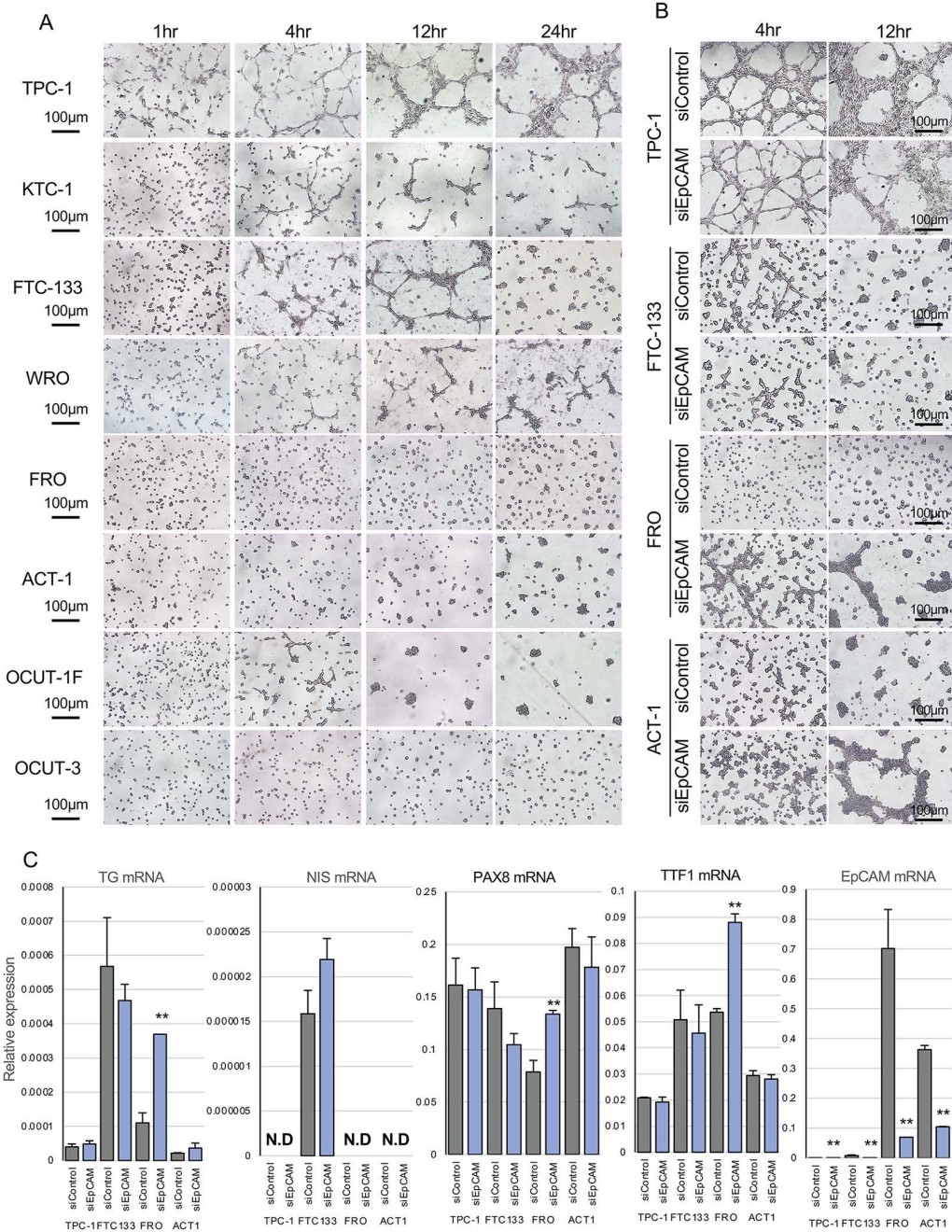

**Fig 3. Effect of EpCAM knockdown on the morphology of thyroid cancer cells on Matrigel. A.** Morphological changes in thyroid cancer cell lines grown on Matrigel-coated plates. Representative images were taken at 1, 4, 12, and 24 h after seeding. **B**. Morphological changes in thyroid cancer cell lines transfected with control or EpCAM siRNA on Matrigel-coated plates. Representative images were taken at 1, 4, 12, and 24 h after seeding. **C**. qRT-PCR analysis showing the mRNA expression levels of thyroglobulin (TG), NIS, PAX8, TTF-1, and EpCAM in Matrigel-coated plates after treatment with EpCAM siRNA for 48 h. **\*\****p* < 0.01.

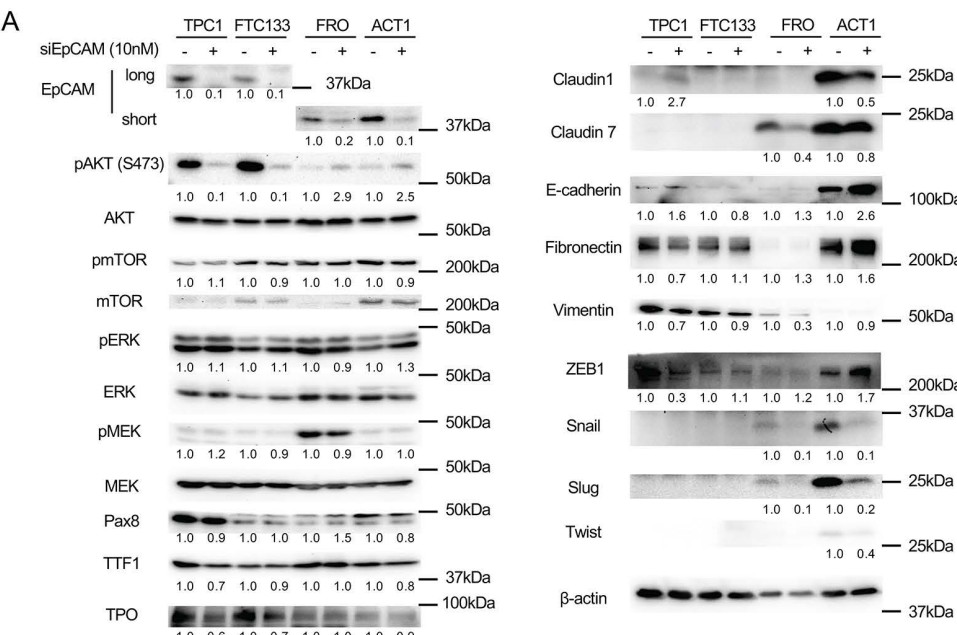

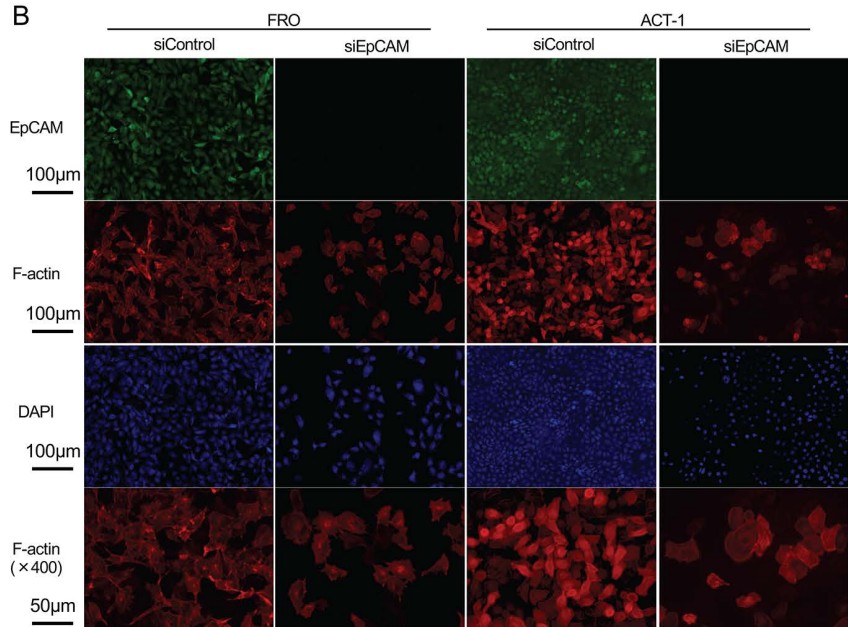

**Fig 4. Effects of EpCAM knockdown on signaling pathways, EMT-related marker expression, and cellular morphology. A.** Western blot analysis showing the expression of cell survival signaling and cell motility-related molecules after treatment with EpCAM siRNA for 48 h. "short" indicates the blot obtained after short exposure time, while "long" indicates results from a long exposure time of the same western blot. Relative protein expression or phosphorylation levels after EpCAM silencing are shown below each band. **B.** Immunofluorescence analysis of F-actin in FRO and ACT-1 cells transfected with control or EpCAM siRNA. Representative images are shown. F-actin was visualized by rhodamine-phalloidin staining (red), and nuclei were stained with DAPI (blue).

FRO and ACT-1 cells, and increased the expression of the epithelial marker E-cadherin in ACT-1 cells. These EMT-related changes were not observed in TPC-1 or FTC-133 cells (Fig 4A).

As cancer cells generally exhibit a more epithelial-like morphology when mesenchymal marker expression is reduced, we examined the morphology of FRO and ACT-1 cells by actin cytoskeleton staining following EpCAM silencing using direct microscopic observation. As shown in Fig 4B, EpCAM knockdown induced clear and reproducible morphological changes in both cell lines, characterized by a transition from a spindle-shaped to a more rounded cell morphology, accompanied by a marked reduction in filopodia formation (Fig 4B).

### EpCAM silencing inhibits extravasation of cancer cells into the lung *in vivo*

Given that EpCAM induced cell motility and filopodia formation *in vitro* (Figs 1–4), we evaluated the effect of EpCAM knockdown on the extravasation of cancer cells into the lung *in vivo*. We examined whether EpCAM promotes extravasation of cancer cells into the lung parenchyma using a short-term tail vein injection experiment (Fig 5A). Based on the fluorescence signal of ACT-1/FITC determined by FACS analysis, we found that the number of infiltrated cancer cells was significantly decreased in the lungs of mice injected with EpCAM siRNA-transfected ACT-1 cells compared with those of mice injected with control siRNA-transfected ACT-1 cells (Fig 5B). Frozen sections were used to visualize cancer cells present in the lungs at 24 h, and consistent with the FACS analysis, we observed that the number of extravasated cells was significantly reduced following EpCAM knockdown (Fig 5C).

### Discussion

We previously demonstrated that EpCAM, together with CD44 and claudin-7, is associated with the phenotype of thyroid cancer in established cell lines and clinical specimens [11]. In addition, we reported that the nuclear expression of EpCAM was significantly increased in ATC compared with DTC in the analysis of clinical thyroid cancer specimens [11]. Similarly, several studies using clinical thyroid cancer specimens have reported that EpCAM expression is an indicator of poor prognosis and high malignancy [26,27]. However, most studies of EpCAM in thyroid cancer biology have focused on evaluating its expression in clinical thyroid cancer tissues by immunohistochemistry [11,26,27], and the mechanisms by which EpCAM is involved in thyroid cancer biology have not been fully elucidated. In the present study, we demonstrated some molecular mechanisms underlying EpCAM-mediated dedifferentiation and increased motility of thyroid cancer cells.

EpCAM has been shown to play versatile roles that extend beyond cell adhesion and include diverse processes such as signaling, cell migration, proliferation, differentiation, metastasis, and drug resistance in several cancer types, and it is known to be overexpressed in many aggressive tumors. Mechanistically, EpCAM mediates the disruption of normal cell–cell junctions, allowing tumor cells to detach and invade surrounding tissues. Furthermore, nuclear translocation of the EpCAM intracellular domain activates oncogenic signaling cascades, such as the Wnt/β-catenin and c-Myc pathways, thereby enhancing proliferation and stemness [9].

Previous studies have demonstrated that increased EpCAM expression significantly correlates with poor prognosis in several cancers [15–17]. Ni et al. reported that EpCAM knockdown significantly inhibited prostate tumor growth and increased sensitivity to chemotherapy/radiotherapy by downregulating the PI3K/AKT/mTOR pathway, resulting in prolonged survival of tumor-bearing mice [28]. In addition, the extracellular domain of EpCAM activates EGFR/ERK signaling through the activation of ADAM17 and γ-secretase [29,30]. These findings suggest that increased EpCAM expression is closely associated with activation of the PI3K/AKT/mTOR and ERK pathways, leading to enhanced cellular proliferation or the development of drug resistance. Consistent with these findings, the present study demonstrated that EpCAM knockdown suppressed AKT phosphorylation in the DTC cell lines TPC-1 and FTC-133. However, EpCAM knockdown did not affect the PI3K/AKT/mTOR or mitogen-activated protein kinase pathway in the ATC cell lines FRO and ACT-1. In ACT-1 and FRO cells, EpCAM knockdown decreased the expression levels of the EMT markers Snail and Slug. In these ATC cell

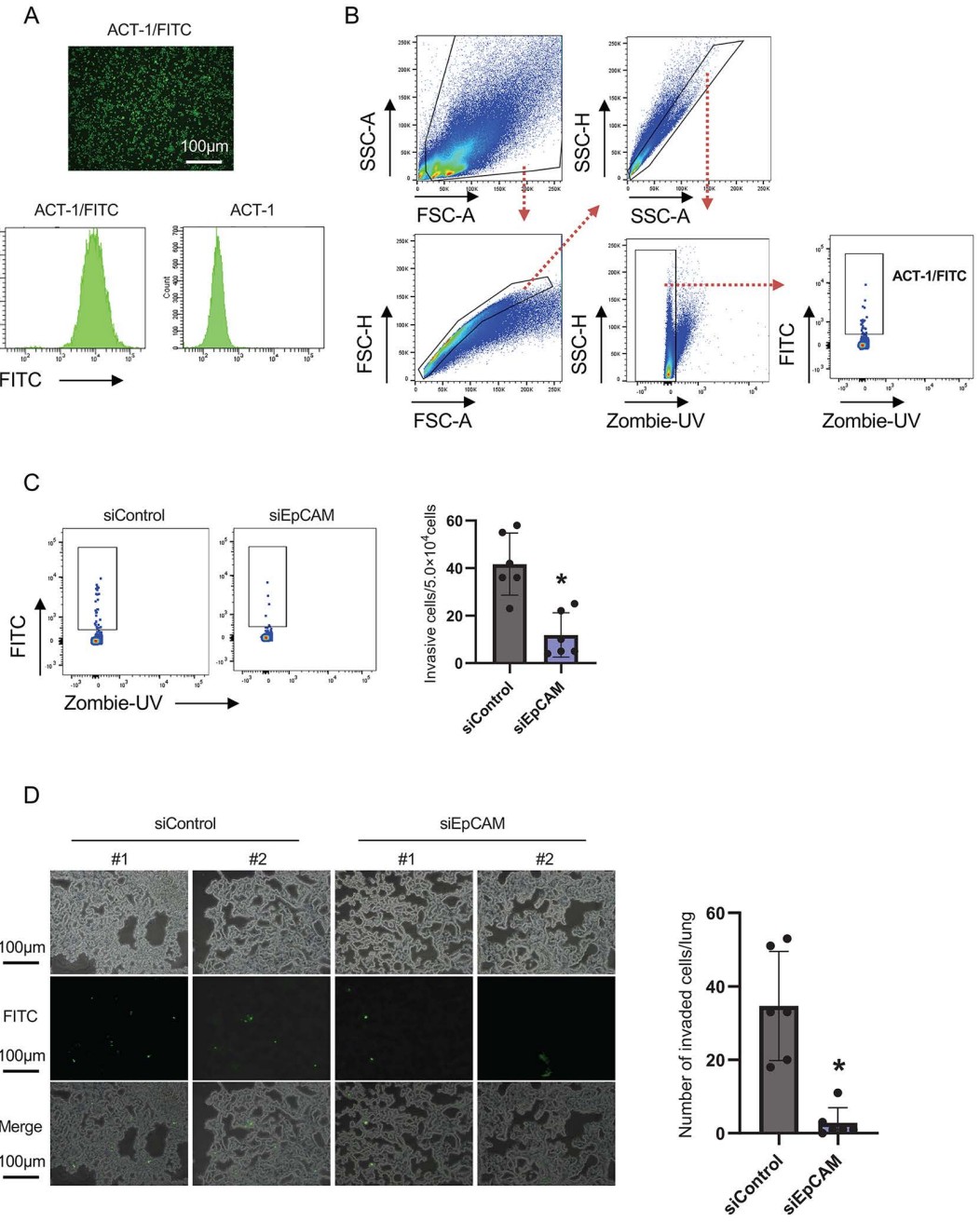

**Fig 5. EpCAM enhances ATC cell extravasation into the lung parenchyma. A.** Fluorescence signal from Green Cell Tracker–labeled cancer cells. Fluorescence intensity was assessed using flow cytometry. **B.** Representative gating strategy for identifying ACT-1/FITC in the lung using flow cytometry. **C.** Representative (left panel) and quantitative (right panel) flow cytometric analyses of cancer cells in the lungs after perfusion with PBS to flush out cancer cells remaining in the circulation 24 h after tail vein injection. *$p < 0.01$. **D**. Representative images of thin lung sections obtained 24 h post-injection (left panel). The number of cancer cell colonies per field is shown in the right panel. siControl, n = 6; siEpCAM, n = 6. *$p < 0.01$.

lines, EpCAM may be involved in cell proliferation through EMT-related molecules. However, further studies are required to elucidate how EpCAM induces the proliferation of these ATC cell lines.

In the present study, the ATC cell lines ATC-1 and FRO exhibited concurrent expression of epithelial and mesenchymal markers. This finding is consistent with previous reports indicating that ATC frequently displays epithelial–mesenchymal plasticity or partial/hybrid EMT, reflecting its dedifferentiated and highly plastic phenotype [31–33].

Regarding the association between EpCAM and metastasis, EpCAM has been reported to be expressed in disseminated and circulating tumor cells (CTCs) [24,34–36] and is used as a diagnostic marker for CTCs in breast, prostate, and colorectal cancers [37]. In thyroid cancer, Lin et al. reported that the number of EpCAM-positive CTCs was associated with poor prognosis in patients who underwent surgery and radioactive iodine therapy for DTC [38]. In the present study, EpCAM silencing in ATC cell lines decreased the expression levels of mesenchymal markers and inhibited lung metastasis formation in mouse models. These results suggest that EpCAM promotes cell migration and invasion via EMT and filopodia formation, thereby contributing to the high metastatic potential of ATC cells observed in patients.

In this study, we obtained results indicating that EpCAM is involved in the proliferation and metastasis of thyroid cancer. Recently, Ghiandai et al. demonstrated, using a three-dimensional spheroid *in vitro* model, that EpCAM is a multifunctional molecule essential for both cell adhesion and intracellular signaling in thyroid cancer cells and may play a central role in cancer progression and stem cell maintenance [39]. These findings emphasize the potential of EpCAM as a promising target for treatment as well as a marker of cancer aggressiveness. To date, the EpCAM monoclonal antibody adecatumumab, the trifunctional antibody catumaxomab (anti-EpCAM and anti-CD3), and the single chain variable fragment of an EpCAM monoclonal antibody, oportuzumab monatox, have demonstrated promising clinical benefits in patients with EpCAM-expressing solid malignancies [40–43]. In addition, the antitumor activity of monoclonal antibodies EpMab-16 and EpMab-37 has been reported in murine xenograft models [44,45]. These findings suggest the feasibility of EpCAM-targeted therapy using antibody-based approaches. Furthermore, EpCAM-directed CAR-T therapy is under development for solid tumors [46]. We have previously shown a high frequency of nuclear EpCAM expression in clinical ATC samples. Novel therapeutic strategies targeting EpCAM may be clinically applied to ATC, although further *in vitro* and *in vivo* studies are warranted.

In this study, we found that EpCAM silencing was associated with increased expression of thyroid differentiation markers in ATC cell lines with high EpCAM expression. Previous studies have reported that EpCAM expression is suppressed during differentiation in undifferentiated pluripotent embryonic stem cells, endodermal progenitor cells, and cancer stem cells, suggesting a role for EpCAM in the regulation of cellular differentiation processes [47–49]. Taken together, our findings indicate that EpCAM expression is linked to the differentiation status of thyroid cancer cells and that EpCAM-expressing ATC cells exhibit vulnerability to EpCAM inhibition. While these results do not establish a direct causal role for EpCAM in thyroid cancer dedifferentiation, they suggest that EpCAM may contribute to the maintenance of aggressive cellular features in ATC.

Although this study presents important findings, several limitations must be considered. We did not verify whether increased EpCAM expression enhances the aggressiveness of thyroid cancer cells. In addition, when analyzing EpCAM function, it is essential to evaluate its intracellular localization, because EpCAM cleavage and intracellular distribution, particularly the nuclear translocation of EpICD, are associated with functional changes induced by EpCAM. However, due to technical limitations, we could not verify the localization of EpCAM in the present *in vitro* EpCAM silencing experiments. We plan to introduce the EpCAM gene into thyroid cancer cells with low EpCAM expression to assess the biological changes induced by increased EpCAM expression in thyroid cancer cells and verify the association between EpICD nuclear translocation and alterations in cellular function. Furthermore, we plan to evaluate whether antibodies that specifically inhibit EpCAM function demonstrate antitumor effects and suppress metastasis using mouse xenograft models *in vivo*.

## Conclusion

To the best of our knowledge, this study provides new evidence that EpCAM is functionally involved in aggressive phenotypic features of ATC cells. Our findings indicate that EpCAM contributes to cell motility, adhesion, invasiveness, and EMT-associated characteristics in EpCAM-expressing ATC cell lines, highlighting EpCAM as a potential vulnerability in this subset of aggressive thyroid cancers. Further research is required to elucidate the precise mechanisms underlying the involvement of EpCAM in the transition from indolent DTC to aggressive ATC; nonetheless, EpCAM may have therapeutic potential in highly aggressive ATC.

## Supporting information

**S1 Table. Characteristics of various thyroid cancer cell lines used in this study.**
(DOCX)

**S2 Table. List of antibodies used for western blot analysis.**
(DOCX)

**S3 Table. Raw absorbance data measured in the cell adhesion assay.**
(DOCX)

**S1 File. Uncropped gel images.**
(PDF)

## Acknowledgments

We would like to thank Editage (www.editage.com) for English language editing.

## Author contributions

**Conceptualization:** Teruo Nakamura, Tomohiro Shibata, Ken-ichi Ito.

**Data curation:** Teruo Nakamura, Tomohiro Shibata.

**Formal analysis:** Teruo Nakamura, Tomohiro Shibata.

**Funding acquisition:** Tomohiro Shibata, Ken-ichi Ito.

**Investigation:** Teruo Nakamura, Tomohiro Shibata.

**Methodology:** Teruo Nakamura, Tomohiro Shibata, Ken-ichi Ito.

**Project administration:** Teruo Nakamura, Tomohiro Shibata, Ken-ichi Ito.

**Resources:** Tomohiro Shibata, Ken-ichi Ito.

**Supervision:** Ken-ichi Ito.

**Visualization:** Ken-ichi Ito.

**Writing – original draft:** Teruo Nakamura, Tomohiro Shibata, Ken-ichi Ito.

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
