## [Decision Letter · Decision Letter 0]

9 Nov 2025

Dear Dr. Ito,

Thank you for submitting your manuscript to PLOS ONE. After careful consideration, we feel that it has merit but does not fully meet PLOS ONE’s publication criteria as it currently stands. Therefore, we invite you to submit a revised version of the manuscript that addresses the points raised during the review process.

**Revisions required for acceptance:**

We look forward to receiving your revised manuscript.

Kind regards,

Tomohito Higashi, Ph.D.

Academic Editor

PLOS ONE

Reviewers' comments:

Reviewer's Responses to Questions

**Comments to the Author**

1. Is the manuscript technically sound, and do the data support the conclusions?

Reviewer #1: Partly

Reviewer #2: Partly

2. Has the statistical analysis been performed appropriately and rigorously?

Reviewer #1: Yes

Reviewer #2: No

3. Have the authors made all data underlying the findings in their manuscript fully available?

Reviewer #1: Yes

Reviewer #2: No

4. Is the manuscript presented in an intelligible fashion and written in standard English?

Reviewer #1: Yes

Reviewer #2: Yes

Reviewer #1: In this Manuscript, Nakamura and colleagues establish a role for epithelial cell adhesion molecule (EPCAM) in promoting aggressive cell behaviors associated with Anaplastic thyroid cancer (ATC). Previous studies have shown correlations between increased expression of EPCAM in aggressive ATC. However, using various thyroid cancer cell lines, and standard cell and molecular biology approaches, this study establishes a causal link between EpCAM, cell proliferation, migration, invasion, differentiation, and cancer cell seeding into distant organs. This is an interesting study that includes robust evidence supporting a causal relationship between EpCAM and several phenotypic features of ATC, however, there are a few points that should be clarified before publication of this manuscript.

Specific Comments:

The authors only used a single siRNA sequence targeting EpCAM for all of their experiments. To ensure effects observed are not due to off targeting of siRNAs, the authors should use at least a second EpCAM siRNA sequence for experiments.

The ‘metastasis assay’ was a tail vein injection followed by 24 hours of cancer cell seeding into the lungs. Metastasis is a multistep process involving invasion and dissemination from the primary tumor, intravasation into circulation and extravasation out of circulation into distant tissues, and finally outgrowth into a robust metastatic tumor. Their assay only assessed extravasation of cancer cells into the lung, and their conclusions about the effects of EpCam should be focused on extravasation of cancer cells and less on EpCam’s role in metastasis.

The authors reference a clonogenic assay used in figure 1C, however the experimental methods for this experiment are not included in materials and methods.

In figure 1A, we see that ATC-1 and FRO cells express several EMT transcription factors (ZEB1, Snail, Twist) and other markers for epithelial cells (E-cadherin). While cells don’t typically express mesenchymal transcription factors and epithelial markers at the same time, it is not impossible (i.e. hybrid EMT cells). To confirm that these are indeed hybrid EM cells, immunfluoresence staining and imaging for EMT transcription factor together with epithelial markers would be appreciated.

Figure 1D and 1E lacks error bars/significance values.

In the western blot in Figure 1D, the FRO cell line does not seem to express appreciable levels of EpCAM, which leaves me wondering whether anything was actually knocked down in these cells?

In figure 2A the authors claim that there is a significant decrease in cell adhesion in TPC-1 cells upon EpCAM knockdown. However, TPC-1 cells do not express appreciable levels of EpCAM as shown in the Western Blots in figure 1A and 2A. How do the authors reconcile this?

In figure 3C there is a typo, the X-axis of the graphs read ‘soEpCAM’ when it should read ‘siEpCAM’.

In figure 4A, there is a clear increase in phospho-AKT levels upon EpCAM depletion. However, the authors claim that there is no effect on pAKT. How do the authors reconcile this?

The authors claim in the text that Twist expression is decreased upon EpCAM knockdown in FRO cells, however, Twist is not expressed in control FRO cells (Fig.4A).

The title of figure 4 implies effects on the actin cytoskeleton upon EpCAM knockdown, however, the authors do assess effects of EpCAM on the actin cytoskeleton. There is mention of effects on filopodia in the text, but again, there was no experimental evidence provided for effects on the actin cytoskeleton of filipodia.

Reviewer #2: The study by Nakamura and colleagues suggests that EpCAM expression is associated with the dedifferentiation status of anaplastic thyroid cancer. Silencing EpCAM expression suppresses proliferation, adhesion, motility, and invasion in anaplastic thyroid cancer cell lines, while increasing the expression of classic thyroid differentiation markers such as PAX8 and TG. The manuscript is well written and offers interesting insights with potential translational significance for the treatment of anaplastic thyroid cancer cases expressing high levels of EpCAM. However, I identified several issues that I hope the authors will clarify. Overall, this study would benefit from a major revision. The data highlight an interesting vulnerability in EpCAM-expressing ATC that could have therapeutic potential. I particularly hope the authors address my concerns regarding the Western Blot images, as the quality of some images limits interpretation. Providing complete membrane images in supplementary information is essential for proper assessment.

Title

I feel that the title is somewhat too generic. The study appears to focus on the knockdown of EpCAM in the anaplastic thyroid cancer cell lines FRO and ACT-1, and on the resulting induction of differentiation and loss of aggressive features. I believe a more suitable title could be chosen.

Introduction

Lines 48–51: While the stepwise progression from PTC to ATC through the accumulation of mutations is a widely accepted theory, the literature also reports cases of anaplastic thyroid cancers that arise independently of a concomitant PTC or differentiated component - for example, the study by Capdevila et al. (2018; PMID: 29648575, DOI: 10.1093/annonc/mdy123). I would suggest revising the sentence to reflect that not all ATCs necessarily originate from a pre-existing differentiated component.

Materials and Methods

Lines 73-76: Why did you consider KTC-1, which originate form a poorly differentiated thyroid carcinoma, in the PTC group? Please, specify here the origin of OCUT-1F and OCUT-3 cell lines.

Line 83: The siRNA sequences or assay IDs should be provided by the authors. I also suggest specifying in which wells the transfection was performed, the time before cell detachment, and how the transfected cells were detached and subsequently reseeded for the different experiments.

Details regarding the amount of transfection reagent used and the number of cells seeded would also be valuable. This information could be included in the supplementary material.

Line 98: The authors should indicate the reverse transcription kit employed and specify the amount of cDNA used per well in the qRTPCR analysis.

Line 101: The authors should clarify which method was used for qPCR data analysis.

Line 103: Although the authors refer to a previous publication for methodological details, I would still recommend reporting the amount of protein loaded and the type of gel used, as these details are essential for reproducibility.

Lines 121-124: How was cell morphology identified or quantified? Tools such as CellProfiler could be used to classify and count cells based on morphological differences. Including such an analysis would strengthen the quantitative value of the data.

Line 144-150: The authors describe the sample preparation clearly; however, a more detailed description of the actual flow cytometry experiment would be useful for readers. For example, why was side scatter analysis chosen instead of forward scatter?

In addition, I highly recommend that the authors include a paragraph detailing all statistical analyses performed in this study — specifically, the tools used, the statistical tests applied, and how morphological data were handled.

Results Section

Lines 154-159: summarize also the results about the other proteins evaluated by WB.

Line 208: The title of this section should be revised to: “EpCAM silencing inhibits dedifferentiation of anaplastic thyroid carcinoma cells.”

Lines 213: I do not see follicle-like network structures in the representative image of FTC-133. Please, comment this observation.

Lines 223–224: This statement seems somewhat too strong. To substantiate a direct link between EpCAM and thyroid cancer dedifferentiation, I suggest performing the reverse experiment — transfecting EpCAM into differentiated thyroid cancer cell lines to assess whether this leads to decreased expression of differentiation markers.

At present, the data convincingly show that ATC cell lines with high EpCAM expression are negatively affected by its silencing, which remains an interesting and meaningful finding on its own.

Lines 244–248: Please refer to my comment on line 121 regarding cell morphology analysis.

Lines 257–265: It is unclear why only the ACT-1 cell line was assessed. Including the FRO cell line would further support the hypothesis that a subset of ATCs expressing EpCAM are particularly sensitive to its knockdown.

Figures

General suggestion: I suggest including all complete Western Blot (WB) images with molecular weight markers in their entirety in the supplementary information. This would allow readers to better interpret some of the banding patterns observed.

Figure 1:

- Figure 1A: Molecular weight markers should be added to all WB images. Include graphs related to densitometric data for the proteins analyzed

- Figure 1C, below panel: The EpCAM expression in FRO appears very low, which contrasts sharply with the results shown in Figure 1A. Is EpCAM expression so variable in FRO? Additionally, there is a band immediately below the ACT-1 EpCAM WB — could this be a non-specific signal?

- Figure 1D and 1E, right panels: I do not see any asterisks indicating statistically significant differences in contrast with those reported in the main text.

Figure 2:

- Figure 2B: The color differences for Collagen IV in FRO and ACT-1 are not apparent, in contrast to Laminin I. This could be due to the representative image chosen. I recommend selecting a clearer representative image and providing raw data in the supplementary material for more accurate evaluation.

- Figure 2C, below panel: EpCAM expression in FRO reappears, which contrasts with Figure 1C. This suggests high variability in FRO EpCAM expression - could the authors comment?

Figure 4:

- Figure 4A: Authors should clarify that the numbers below the bands represent fold changes or delete them from the image.

- It might be useful to limit the number of WBs in the main figures and include graphs for the most important proteins, moving WB images to supplementary information.

- The method for assessing pBRAF and pCRAF expression is unclear; total protein expression appears not to have been used for normalization.

- EpCAM expression in TPC-1 and FTC-133 appears higher than in Figure 1A. I strongly recommend that authors show both long and short EpCAM isoforms for all cell lines.

- Snail WB shows a minor band aberration; a clearer representative image is advised.

- Twist WB appears excessively white in TPC-1 and FTC-133; a better representative image with less contrast should be selected.

Figure 5:

- Figures 5A and 5B: The image descriptions should be clarified. Please see my previous comment on flow cytometry in the Materials and Methods section.

Discussion

Lines 331 and 336: These conclusions appear somewhat too strong given the data presented. So far, the authors have demonstrated a vulnerability in EpCAM-expressing ATC cell lines.

Line 341: The current results do not provide strong evidence for this statement. Additional experiments in EpCAM-expressing differentiated thyroid cancer (DTC) would be necessary to support this claim. I recommend removing or rephrasing this sentence.

Conclusion

Lines 355–356: Refer to my comments in the Discussion section.

Line 359: Consider using a synonym for “virulent” - for example, “aggressive” - which may be more appropriate in the context of thyroid cancer.

**Do you want your identity to be public for this peer review?** For information about this choice, including consent withdrawal, please see our Privacy Policy

Reviewer #1: No

Reviewer #2: No

---

## [Author Response · Author response to Decision Letter 1]

17 Jan 2026

Response to Reviewers

We sincerely thank the Academic Editor and the Reviewers for their careful evaluation of our manuscript and for the thoughtful and constructive comments provided. We greatly appreciate the time and effort invested in reviewing our work.

In response to the Reviewers’ comments, we have performed additional experiments, including validation using multiple independent siRNA sequences, and have extensively revised the manuscript to improve its clarity, rigor, and reproducibility. We believe that these additional data and revisions have substantially strengthened the scientific quality of the manuscript and clarified the scope and significance of our findings. In the revised manuscript, all changes have been indicated using red text with yellow highlighting to facilitate review.

We hope that the revised manuscript now adequately addresses all concerns raised by the reviewers, and we respectfully submit the following point-by-point responses to each comment. We sincerely hope that the revised version will be suitable for publication in PLOS One.

Response to Reviewer #1

In this Manuscript, Nakamura and colleagues establish a role for epithelial cell adhesion molecule (EPCAM) in promoting aggressive cell behaviors associated with Anaplastic thyroid cancer (ATC). Previous studies have shown correlations between increased expression of EPCAM in aggressive ATC. However, using various thyroid cancer cell lines, and standard cell and molecular biology approaches, this study establishes a causal link between EpCAM, cell proliferation, migration, invasion, differentiation, and cancer cell seeding into distant organs. This is an interesting study that includes robust evidence supporting a causal relationship between EpCAM and several phenotypic features of ATC, however, there are a few points that should be clarified before publication of this manuscript.

We sincerely thank Reviewer #1 for the thorough and constructive evaluation of our manuscript. We greatly appreciate the insightful comments, which have helped us identify important points requiring clarification and additional experimentation. Below, we provide a point-by-point response to the comments raised. All changes have been incorporated into the revised manuscript.

Comment 1:

The authors only used a single siRNA sequence targeting EpCAM for all of their experiments. To ensure effects observed are not due to off targeting of siRNAs, the authors should use at least a second EpCAM siRNA sequence for experiments.

Response:

We thank the reviewer for this important comment. We performed additional validation experiments using two independent siRNA sequences targeting EpCAM and evaluated EpCAM silencing, cell proliferation, migratory ability, and invasion following EpCAM knockdown in the FRO and ACT-1 anaplastic thyroid cancer cell lines.

Importantly, EpCAM silencing using the additional siRNA yielded results consistent with those originally reported, including comparable suppression of cell proliferation, migratory ability, and invasiveness. These findings confirm that the observed effects are reproducible and unlikely to be attributable to off-target effects of a single siRNA. The new data obtained using multiple siRNA sequences have now been incorporated into the revised manuscript, and the corresponding figures (Figure 1B, 1D, and 1E) have been replaced accordingly.

Comment 2:

The ‘metastasis assay’ was a tail vein injection followed by 24 hours of cancer cell seeding into the lungs. Metastasis is a multistep process involving invasion and dissemination from the primary tumor, intravasation into circulation and extravasation out of circulation into distant tissues, and finally outgrowth into a robust metastatic tumor. Their assay only assessed extravasation of cancer cells into the lung, and their conclusions about the effects of EpCAM should be focused on extravasation of cancer cells and less on EpCAM’s role in metastasis.

Response:

We appreciate this clarification and agree with the reviewer’s point. Our assay specifically evaluates the extravasation of tumor cells into lung tissue rather than the multi-step metastatic cascade. Accordingly, we have revised the terminology throughout the manuscript by replacing “metastasis” with “lung extravasation,” and have toned down any claims implying a full metastatic process.

Comment 3:

The authors reference a clonogenic assay used in Figure 1C, however the experimental methods for this experiment are not included in materials and methods.

Response:

Thank you for pointing this out. We have now added a detailed description of the “Colony formation assay” protocol—including cell seeding density, incubation time, staining procedures, and quantification method—to the subsection in the “Materials and Methods” section.

Comment 4:

In Figure 1A, we see that ATC-1 and FRO cells express several EMT transcription factors (ZEB1, Snail, Twist) and other markers for epithelial cells (E-cadherin). While cells don’t typically express mesenchymal transcription factors and epithelial markers at the same time, it is not impossible (i.e. hybrid EMT cells). To confirm that these are indeed hybrid EM cells, immunfluoresence staining and imaging for EMT transcription factor together with epithelial markers would be appreciated.

Response:

We thank the reviewer for the insightful comment regarding the EMT status of anaplastic thyroid cancer (ATC) cells. ATC is generally considered to arise through dedifferentiation from differentiated thyroid carcinoma, a process associated with marked phenotypic plasticity. Previous studies have shown that this transformation is not necessarily accompanied by a complete epithelial–mesenchymal transition (EMT), but rather by partial or hybrid EMT states, in which epithelial characteristics are altered but not uniformly lost. In this context, derangement of the E-cadherin/catenin complex has been implicated in the transformation from differentiated to anaplastic thyroid carcinoma (Wiseman et al., Am J Surg, 2006), and immunohistochemical analyses have demonstrated expression of EMT regulators such as SNAI2 and TWIST1 in thyroid carcinomas, including ATC (Buehler et al., Mod Pathol, 2013).

In addition, experimental studies using ATC cell lines have reported mixed epithelial–mesenchymal phenotypes. Notably, Baldini et al. demonstrated that the ATC cell line BHT-101 expresses the epithelial marker E-cadherin along with EMT-associated factors including TWIST1 and fibronectin, and interpreted these findings in the context of epithelial–mesenchymal plasticity rather than a complete EMT program (Int J Mol Sci, 2024).

In the present study, Figure 1A is based on immunoblot analyses of bulk cell lysates and was intended to characterize the overall EMT-related expression profile of ATC cell populations. Although immunofluorescence co-staining would be an ideal approach to demonstrate hybrid EMT at the single-cell level, we considered that, in light of existing pathological and experimental evidence supporting epithelial–mesenchymal plasticity in ATC, additional immunofluorescence analyses were not essential for interpretation of the current findings.

To clarify this interpretation for readers, we have added a corresponding discussion to the paragraph 2 on page 15 in the “Discussion” section, emphasizing that ATC-1 and FRO cells exhibit epithelial–mesenchymal plasticity or partial/hybrid EMT rather than a complete EMT phenotype, and citing the relevant pathological and experimental literature.

Comment 5:

Figure 1D and 1E lacks error bars/significance values.

Response:

Thank you very much for pointing out this important issue. In response to the reviewer’s comment, we performed additional experiments using two independent siRNA sequences targeting EpCAM to confirm the reproducibility of the results. In both ACT-1 and FRO cells, EpCAM knockdown consistently led to a significant reduction in cell migratory ability.

We have now incorporated these new data into the revised Figures 1D and 1E. The updated figures include:

• Error bars indicating standard deviation,

• Statistical analyses showing the significance of the observed differences, and

• Biological replicates using multiple siRNAs, confirming reproducibility.

The original panels have been replaced with these new results, and the figure legends have been updated accordingly.

Comment 6:

In Figure 2A the authors claim that there is a significant decrease in cell adhesion in TPC-1 cells upon EpCAM knockdown. However, TPC-1 cells do not express appreciable levels of EpCAM as shown in the Western Blots in figure 1A and 2A. How do the authors reconcile this?

Response:

We appreciate this important question. As shown in the Western blot analyses (Figures 1A and 2A), EpCAM expression in TPC-1 cells is lower than that observed in ATC cell lines such as ACT-1 and FRO, but is not completely absent. Accordingly, EpCAM knockdown in TPC-1 cells resulted in a modest but statistically significant reduction in cell adhesion. Notably, the magnitude of this effect was considerably smaller than that observed in EpCAM-high ATC cell lines.

In contrast, EpCAM knockdown did not significantly affect Matrigel adhesion in FTC-133 cells. This may reflect differences in intracellular signaling context, as FTC-133 cells harbor a PTEN mutation and exhibit constitutive activation of the PI3K/AKT pathway. In addition, cell adhesion in FTC-133 cells may be regulated predominantly by β-catenin–dependent or alternative adhesion-related pathways, thereby rendering them less dependent on EpCAM-mediated adhesion mechanisms. Although this interpretation remains speculative, we have revised the “Results” and “Discussion” sections to acknowledge these potential mechanisms and to avoid overinterpretation.

Comment 7:

In figure 3C there is a typo, the X-axis of the graphs read ‘soEpCAM’ when it should read ‘siEpCAM’.

Response:

We thank the reviewer for pointing out this typographical error. The labeling has been corrected from “soEpCAM” to “siEpCAM” in Figure 3C of the revised manuscript.

Comment 8:

In figure 4A, there is a clear increase in phospho-AKT levels upon EpCAM depletion. However, the authors claim that there is no effect on pAKT. How do the authors reconcile this?

The authors claim in the text that Twist expression is decreased upon EpCAM knockdown in FRO cells, however, Twist is not expressed in control FRO cells (Fig.4A).

Response:

We thank the reviewer for these important observations. We agree that the original description did not adequately reflect the experimental data shown in Figure 4A, including both AKT phosphorylation and EMT marker expression. Accordingly, we have revised the “Results” section to accurately describe these findings.

Regarding AKT signaling, we now state that EpCAM knockdown caused a marked reduction in AKT phosphorylation in TPC-1 and FTC-133 cells (approximately tenfold), whereas a modest increase in phospho-AKT was observed in the ATC cell lines FRO and ACT-1. Importantly, the increase in AKT phosphorylation in FRO and ACT-1 cells was considerably smaller than the pronounced decrease observed in TPC-1 and FTC-133 cells, and was not accompanied by consistent changes in other major cell growth–related signaling molecules.

In addition, we corrected the description of EMT-related markers to address the reviewer’s concern regarding Twist expression. As Twist was not detectable in control FRO cells, we have removed this marker from the description of EpCAM knockdown effects in FRO cells. The revised “Results” section now states that EpCAM knockdown decreased the expression of the mesenchymal markers Snail and Slug in FRO and ACT-1 cells, and increased the expression of the epithelial marker E-cadherin in ACT-1 cells, while no such EMT-related changes were observed in TPC-1 or FTC-133 cells (paragraph 1 on page 12).

Comment 9:

The title of figure 4 implies effects on the actin cytoskeleton upon EpCAM knockdown, however, the authors do assess effects of EpCAM on the actin cytoskeleton. There is mention of effects on filopodia in the text, but again, there was no experimental evidence provided for effects on the actin cytoskeleton of filipodia.

Response:

We thank the reviewer for this helpful comment. To address this concern, we have revised the title of Figure 4 to accurately reflect the data presented as follows:

“Fig. 4. Effects of EpCAM knockdown on signaling pathways, EMT-related marker expression, and cellular morphology.”

Response to Reviewer #2

The study by Nakamura and colleagues suggests that EpCAM expression is associated with the dedifferentiation status of anaplastic thyroid cancer. Silencing EpCAM expression suppresses proliferation, adhesion, motility, and invasion in anaplastic thyroid cancer cell lines, while increasing the expression of classic thyroid differentiation markers such as PAX8 and TG. The manuscript is well written and offers interesting insights with potential translational significance for the treatment of anaplastic thyroid cancer cases expressing high levels of EpCAM. However, I identified several issues that I hope the authors will clarify. Overall, this study would benefit from a major revision. The data highlight an interesting vulnerability in EpCAM-expressing ATC that could have therapeutic potential.

Comment 1:

I particularly hope the authors address my concerns regarding the Western blot images, as the quality of some images limits interpretation. Providing complete membrane images in supplementary information is essential for proper assessment.

Response:

We thank the reviewer for this important comment. In response to this concern, we have now provided complete, uncropped western blot membrane images for all blots presented in the manuscript as “Figure S” in the “Supplementary information.” These original images include molecular weight markers and are provided without adjustment to ensure transparency and allow proper assessment of the data. We believe that the inclusion of these uncropped membrane images addresses the reviewer’s concern and improves the rigor and reproducibility of the study.

Comment 2:

Title:

I feel that the title is somewhat too generic. The study appears to focus on the knockdown of EpCAM in the anaplastic thyroid cancer cell lines FRO and ACT-1, and on the resulting induction of differentiation and loss of aggressive features. I believe a more suitable title could be chosen.

Response:

We thank the reviewer for this helpful suggestion. In response, we have revised the manuscript title to more accurately reflect the focus of this study on EpCAM knockdown in anaplastic thyroid cancer cells and its effects on aggressive behavior and differentiation. The revised title is as follows:

“EpCAM silencing suppresses aggressive phenotypes and induces partial redifferentiation in anaplastic thyroid cancer cells”

In addition, we have revised the short title to better align with the revised main title while avoiding overstatement:

Short title: “EpCAM and aggressive phenotypes in anaplastic thyroid cancer”

Introduction:

Comment 3:

Lines 48–51: While the stepwise progression from PTC to ATC through the accumulation of mutations is a widely accepted theory, the literature also reports cases of anaplastic thyroid cancers that arise independently of a concomitant PTC or differentiated component - for example, the study by Capdevila et al. (2018; PMID: 29648575, DOI: 10.1093/annonc/mdy123). I would suggest revising the sentence to reflect that not all ATCs necessarily originate from a pre-existing differentiated component.

Response:

We thank the reviewer for this important and insightful comment. We agree that the original description placed excessive emphasis on a linear progression model from differentiated thyroid cancer to anaplastic thyroid cancer. Accordingly, we have revised the “Introduction” to clarify that not all ATCs necessarily arise from a pre-existing differentiated component, and that ATC and PTC may diverge early during tumor evolution and evolve as molecularly distinct entities (paragraph 2 on page 3). We have incorporated and cited the study by Capdevila et al. (Ann Oncol, 2018) to support this revised statement.

Materials and Methods:

Commen

---

## [Decision Letter · Decision Letter 1]

5 Feb 2026

Dear Dr. Ito,

We look forward to receiving your revised manuscript.

Kind regards,

Tomohito Higashi, Ph.D.

Academic Editor

PLOS One

Journal Requirements:

Reviewers' comments:

Reviewer's Responses to Questions

**Comments to the Author**

Reviewer #1: All comments have been addressed

Reviewer #2: (No Response)

2. Is the manuscript technically sound, and do the data support the conclusions?

Reviewer #1: Yes

Reviewer #2: Partly

3. Has the statistical analysis been performed appropriately and rigorously?

Reviewer #1: Yes

Reviewer #2: No

4. Have the authors made all data underlying the findings in their manuscript fully available?

Reviewer #1: Yes

Reviewer #2: No

5. Is the manuscript presented in an intelligible fashion and written in standard English?

Reviewer #1: Yes

Reviewer #2: Yes

Reviewer #1: All of my comments were addressed to my satisfaction. I believe the revisions have significantly increased the quality of the manuscript.

Reviewer #2: I thank the authors for satisfactorily addressing all the comments raised. I still have one concern that needs to be expressed regarding Figure 1A, specifically the lack of a densitometric analysis. As stated by the authors, this experiment shows that only a limited subset of thyroid cancer cell lines expresses EpCAM. While this is clearly visible in the Western blot images, I believe that a densitometric analysis (based on data obtained in triplicate) would provide much stronger evidence than qualitative images alone, which could then be moved to the Supplementary Material. The β-actin in Figure 1 appears to be overexposed. Could the authors please provide a more representative image?

**Do you want your identity to be public for this peer review?** For information about this choice, including consent withdrawal, please see our Privacy Policy

Reviewer #1: No

Reviewer #2: No

---

## [Author Response · Author response to Decision Letter 2]

13 Feb 2026

Response to Reviewers

We sincerely thank the Academic Editor and Reviewers for their careful re-evaluation of our revised manuscript and their thoughtful and constructive comments. We are grateful for Reviewer #1’s positive assessment of the revisions, and below we provide our response to Reviewer #2’s additional comments regarding Figure 1A.

Response to Reviewer #2

Reviewer #2: I thank the authors for satisfactorily addressing all the comments raised. I still have one concern that needs to be expressed regarding Figure 1A, specifically the lack of a densitometric analysis. As stated by the authors, this experiment shows that only a limited subset of thyroid cancer cell lines expresses EpCAM. While this is clearly visible in the Western blot images, I believe that a densitometric analysis (based on data obtained in triplicate) would provide much stronger evidence than qualitative images alone, which could then be moved to the Supplementary Material. The β-actin in Figure 1 appears to be overexposed. Could the authors please provide a more representative image?

Response:

We thank the reviewer for this constructive suggestion. In response, we have now performed densitometric quantification of EpCAM expression based on three independent western blot experiments and added the resulting bar graphs to Figure 1A. These quantitative analyses demonstrated that EpCAM expression was significantly higher in FRO and ACT-1 cells than in the other thyroid cancer cell lines. To ensure full transparency, the corresponding uncropped gel images used for quantification are now provided in the Supplementary Information (Figure S-Uncropped images for EpCAM in Figure 1A).

In addition, to improve the presentation of the loading controls, we have now shown α-tubulin, which was used as a loading control in the same experiments. As an appropriately exposed β-actin image was not available for these particular blots, we elected to present the α-tubulin data obtained from the same membranes instead in the revised figure.

Consistent with these new quantitative results, we have revised the Results section to explicitly state that FRO and ACT-1 cells express EpCAM at significantly higher levels than the other thyroid cancer cell lines, and that these two cell lines were therefore used primarily in subsequent functional experiments. Figure 1A legend has been updated accordingly to describe densitometric analysis.

We believe that these additions and revisions fully address the reviewer’s concerns and further strengthen the rigor of the manuscript.

---

## [Editor Report · Decision Letter 2]

19 Feb 2026

EpCAM silencing suppresses aggressive phenotypes and induces partial redifferentiation in anaplastic thyroid cancer cells.

PONE-D-25-40514R2

Dear Dr. Ito,

We’re pleased to inform you that your manuscript has been judged scientifically suitable for publication and will be formally accepted for publication once it meets all outstanding technical requirements.

Kind regards,

Tomohito Higashi, Ph.D.

Academic Editor

PLOS One
---

## [Editor Report · Acceptance letter]

PONE-D-25-40514R2

PLOS One

Dear Dr. Ito,

I'm pleased to inform you that your manuscript has been deemed suitable for publication in PLOS One. Congratulations! Your manuscript is now being handed over to our production team.

Kind regards,

on behalf of

Dr. Tomohito Higashi

Academic Editor

PLOS One